# Signal Detection Based on Power-Spectrum Sub-Band Energy Ratio

**Han Li** [1,2] , **Yanzhu Hu** [1,*] **and Song Wang** [1]

1   School of Modern Post, Beijing University of Posts and Telecommunications, Beijing 100876, China; lihan_60@bupt.edu.cn (H.L.); wongsang@bupt.edu.cn (S.W.)
2   School of Intelligent Equipment, Shandong University of Science and Technology, Tai'an 271019, China
*   Correspondence: bupt_automation_safety_yzhu@bupt.edu.cn; Tel.: +86-010-133-6603-6076

**Abstract:** The power-spectrum sub-band energy ratio (PSER) has been applied in a variety of fields, but reports on its statistical properties and application in signal detection have been limited. Therefore, the statistical characteristics of the PSER were investigated and a signal detection method based on the PSER was created in this paper. By analyzing the probability and independence of power spectrum bins, as well as the relationship between F and beta distributions, we developed a probability distribution for the PSER. Our results showed that in a case of pure noise, the PSER follows beta distribution. In addition, the probability density function exhibited no relationship with the noise variance—only with the number of bins in the power spectrum. When Gaussian white noise was mixed with the signal, the resulting PSER followed a doubly non-central beta distribution. In this case, the probability density and cumulative distribution functions were represented by infinite double series. Under the constant false alarm strategy, we established a signal detector based on the PSER and derived the false alarm probability and detection probability of the PSER. The main advantage of this detector is that it did not need to estimate noise variance. Compared with time-domain energy detection and local spectral energy detection, we found that the PSER had better robustness under noise uncertainty. Finally, the results in the simulation and real signal showed that this detection method was valid.

**Keywords:** power-spectrum sub-band energy ratio; beta distribution; doubly non-central beta distribution; infinite double series; noise uncertainty

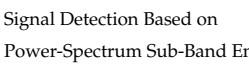



## 1. Introduction

The power-spectrum sub-band energy ratio (PSER) is a common metric used to represent the proportion of signal energy in a single spectral line. The PSER is derived from spectral analysis and has a direct relation with the amplitude and power spectra. The signal mixed with additive Gaussian white noise (GWN) can be expressed as

$$s(n) = \begin{cases} z(n) & H_0 \\ x(n) + z(n) & H_1 \end{cases}, \tag{1}$$

where $n = 0, 1, \cdots, N-1$; $s(n)$ is the signal to be detected; $x(n)$ is the signal; $z(n)$ is the GWN with a mean of zero and a variance of $\sigma^2$; $H_0$ represents the hypothesis corresponding to "no signal transmitted;" and $H_1$ corresponds to "signal transmitted."

The spectrum of $s(n)$ is given by

$$\overrightarrow{S}(k) = \sum_{n=0}^{N-1} s(n) e^{-j\frac{2\pi}{N}kn}, k = 0, 1, \cdots, N-1. \tag{2}$$

where $N$ is the number of spectral bins, $j$ is the imaginary unit, and the arrow superscript denotes a complex function. The $k$th line in the power spectrum of $s(n)$ can be expressed as

$$P(k) = \frac{1}{N} \left| \vec{S}(k) \right|^2 . \tag{3}$$

The PSER $B_{d,N}(k)$ is defined as the ratio of the sum of adjacent $d$ bins from the $k$th bin in the power spectrum to the entire spectrum energy, i.e.,

$$B_{d,N}(k) = \frac{\sum_{l=0}^{d-1} P(kd+l)}{\sum_{i=0}^{N-1} P(i)}, 1 \leq d < N, \ k = 0, 1, \cdots, N'-1 , \tag{4}$$

where $\sum_{i=0}^{N-1} P(i)$ represents the total energy in the power spectrum and $\sum_{l=0}^{d-1} P(kd+l)$ represents the total energy of adjacent $d$ bins. There are $N$ spectral bins in the power spectrum, and $\lfloor N/d \rfloor$ sub-bands are formed after the combination. The symbol $\lfloor \ \rfloor$ stands for rounding down. Let $N' = \lfloor N/d \rfloor$, then $k = 0, 1, \cdots, N'-1$.

The PSER is consistent with the spectrum in the waveform, and the data dimension can be greatly reduced by combining spectral bins. Therefore, it has been extensively applied in the fields of remote communication [1,2], earthquake modeling [3], machine design [4], and geological engineering [5]. For example, Preisig et al. used the PSER to study signal detection in underwater acoustic environments [6]. Wang et al. described the distribution of acoustic emission signals in the fracturing of fine sandstone [7]. Xu and Guan extracted the characteristics of underground micro-seismic signals collected by a distributed optical fiber using the PSER [8]. Huang et al. investigated the effects of various dual-peak spectral parameters on dispersive coefficients [9]. Yang et al. used the PSER as a signal feature in coal gangue recognition [10]. Lu et al. studied micro-seismic signals in dual-layer igneous strata [11]. Kong et al. described the feature of electromagnetic radiation under coal oxidation [12]. Though the PSER is widely used, research on its statistical characteristics and detection performance is not sufficient and must be further improved.

Statistical characteristics are the theoretical basics of signal detection. The statistical characteristics of frequency and power spectra have been extensively studied for GWN. The real and imaginary parts of the spectrum follow a Gaussian distribution [2], and the amplitude conforms to a Rayleigh distribution [13]. Groth analyzed the statistical characteristics of the power spectrum with white noise [14]. Johnson et al. studied the distribution characteristics of power spectra acquired by the periodogram averaging technique [15]. Martinez et al. presented a detector based on the Welch's periodogram [16], and the test statistic of this method was approximated by the Gaussian distribution. Pei-Han proposed a power spectral density split cancellation with the same definition as the PSER and derived its probability density function (PDF) and cumulative distribution function (CDF); however, its probability distribution was not specified [2,17]. Bomfin et al. proposed the cooperative power spectral density split cancellation method [18], which was more robust against noise uncertainty than the method of Pei-Han. Bomfin et al. proposed circular folding cooperative power spectral density split cancellation and derived a closed form expression for false alarm threshold [19]. However, he did not provide the expression for CDF when a signal presented. Gurugopinath et al. proposed the fast Fourier transform average ratio (FAR) algorithm that was the same as the PSER. The CDF for the FAR was expressed by a hypergeometric function [20], which was more complicated than our method. In this study, we qualitatively analyzed the probability distribution of the PSER and obtained the accurate PDF and CDF of different existing formulas.

The purpose of signal detection is to detect whether there is a signal on the sampling sequence under a noisy environment. Common signal detection methods [21,22] can be classified as matched filter detection [23], energy-based detection [24], and cyclostationary-based detection [25], among others. Energy detection can be classified into two categories: time domain energy detection and frequency domain energy detection. The PSER belongs to a frequency-domain energy detection technology. According to the statistical character-

istics of the PSER, we deduced the signal detection method based on the PSER under the constant false alarm (CFAR) strategy.

The remainder of this paper is organized as follows. Section 2 discusses the statistical characteristics of the PSER for GWN. Section 3 develops the signal detection method based on the PSER under the CFAR. In Section 4, simulation experiments with the narrowband and wideband signals are separately used to verify the accuracy of the PSER detection method. The application in vibration signals suggests that the PSER detector can work well in real environments. Section 5 provides additional details concerning the research process.

## 2. Statistical Characteristics of the PSER

### 2.1. Statistical Characteristics of GWN

The GWN spectrum, calculated using a discrete Fourier transform (DFT), is given by

$$\vec{Z}(k) = \sum_{n=0}^{N-1} z(n)[\cos(\frac{2\pi}{N}kn) - j\sin(\frac{2\pi}{N}kn)], \ k = 0, 1, 2, \cdots, N-1 \,. \tag{5}$$

Let the real and imaginary parts of $\vec{Z}(k)$ be $Z_R(k)$ and $Z_I(k)$, respectively. Then $Z_R(k)$ and $Z_I(k)$ are mutually independent, and they all follow a Gaussian distribution [14]; that is,

$$Z_R(k) \sim \mathcal{N}(0, \frac{N\sigma^2}{2}), Z_I(k) \sim \mathcal{N}(0, \frac{N\sigma^2}{2}) \,. \tag{6}$$

The power spectrum for the term $\vec{Z}(k)$ is defined as

$$P_Z(k) = \frac{1}{N}\left|\vec{Z}(k)\right|^2 = \frac{1}{N}\left[Z_R{}^2(k) + Z_I{}^2(k)\right], \ k = 0, 1, \cdots, N-1 \,. \tag{7}$$

$P_Z(k)$ is also called a power spectrum bin. Any two white noise power spectral bins are independent [2], and $2P_z(k)/\sigma^2$ follows a chi-squared distribution with two degrees of freedom [2].

### 2.2. Statistical Characteristics of Power Spectrum Bin for GWN

2.2.1. Statistical Characteristics of One Power Spectrum Bin

Under $H_1$, the discrete Fourier transform of $s(n)$ can be expressed as

$$\vec{S}(k) = \vec{X}(k) + \vec{Z}(k), \ k = 0, 1, \cdots, N-1 \,. \tag{8}$$

The power spectra for $\vec{X}(k)$ and $\vec{Z}(k)$ are then given by, respectively,

$$P_X(k) = \frac{1}{N}\left[X_R^2(k) + X_I^2(k)\right], \ k = 0, 1, \cdots, N-1 \,, \tag{9}$$

$$P_Z(k) = \frac{1}{N}\left[Z_R^2(k) + Z_I^2(k)\right], \ k = 0, 1, \cdots, N-1 \,. \tag{10}$$

The power spectrum for $\vec{S}(k)$ is

$$P_S(k) = \frac{1}{N}\left\{[X_R(k) + Z_R(k)]^2 + [X_I(k) + Z_I(k)]^2\right\}, k = 0, 1, \cdots, N-1 \,. \tag{11}$$

Let $S_R(k) = X_R(k) + Z_R(k)$ and $S_I(k) = X_I(k) + Z_I(k)$. Since $\vec{X}(k)$ represents the spectrum of a known signal, the real and imaginary parts of the spectrum can be assumed to be constant (i.e., $a_k = X_R(k)$ and $b_k = X_I(k)$) [14]. As a result,

$$S_R(k) \sim \mathcal{N}(a_k, \frac{N\sigma^2}{2}), \ S_I(k) \sim \mathcal{N}(b_k, \frac{N\sigma^2}{2}) \,. \tag{12}$$

#### 2.2.2. Statistical Characteristics of Sum of Multiple Power Spectra Bins

For the convenience of description, the power spectrum bin $P(i)$ in the next section is replaced by the random variable $X_k$ (i.e., $X_k = P(k)$). The sum of $d$ power spectrum bins, $\sum_{l=0}^{d-1} P(kd+l)$, is replaced by the random variable $X'_k$, i.e.,

$$X'_k = P(kd) + P(kd+1)\cdots P(kd+d-1) . \tag{13}$$

Let $\lambda_i = 2\left(a_i^2 + b_i^2\right)/\left(N\sigma^2\right)$ and $\lambda'_k = \sum_{l=0}^{d-1} \lambda_{k+l}$. It can be shown that $2X'_k/\sigma^2$ follows a non-central chi-square distribution with $d$ degrees of freedom and a non-centrality parameter $\lambda'_k$ [26]. As such,

$$\frac{2}{\sigma^2} X'_k \sim \chi_{2d}^2\left(\lambda'_k\right) . \tag{14}$$

#### 2.3. Identifying Probability Distribution for the PSER

The numerator and denominator of $B_{d,N}(k)$ are not independent, and the probability distribution for $B_{d,N}(k)$ cannot be calculated from the probability distribution for $X'_k$ and $\sum_{i=0}^{N-1} X_i$.

The random variable $B'_{d,N}(k)$ represents the ratio of $X'_k$ to the sum of the remaining $N-d$ variables that do not contain $X'_k$, i.e.,

$$B'_{d,N}(k) = \frac{X'_k}{\sum_{i=0}^{N-1} X_i - X'_k} , k = 0, 1, \cdots, N'-1 . \tag{15}$$

$B'_{d,N}(k)$ spans a range of $[0, +\infty)$. Note that the numerator $B'_{d,N}(k)$ is not included in the denominator, thus indicating the numerator and denominator of $B'_{d,N}(k)$ are independent. The probability distribution for $B'_{d,N}(k)$ can then be calculated from the probability distributions of $X'_k$ and $\sum_{i=0}^{N-1} X_i - X'_k$. As a result, $B_{d,N}(k)$ and $B'_{d,N}(k)$ exhibit the following relationship:

$$\begin{aligned} F_{B_{d,N}(k)}(y) &= \Pr\left(\frac{X'_k}{\sum_{i=0}^{N-1} X_i} < y\right) \\ &= \Pr\left(\frac{X'_k}{\sum_{i=0}^{N-1} X_i - X'_k} < \frac{y}{1-y}\right) \\ &= F_{B'_{d,N}(k)}\left(\frac{y}{1-y}\right) \qquad , 0 < y < 1, N \geq d+1 . \end{aligned} \tag{16}$$

Using this equation, the distribution of $B_{d,N}(k)$ can be calculated indirectly from the distribution of $B'_{d,N}(k)$.

#### 2.4. Statistical Characteristics of the PSER under $H_0$

##### 2.4.1. Probability Distribution for $B'_{d,N}(k)$

The numerator $X'_k$ in $B'_{d,N}(k)$ follows a chi-squared distribution with $2d$ degrees of freedom, and the denominator $\sum_{i=0}^{N-1} X_i - X'$ follows the same distribution with $2N-2d$ degrees of freedom. The product $(N-d)B'_{d,N}(k)/d$ follows an $F$ distribution$-$ with $2d$, $2N-2d$ degrees of freedom:

$$\frac{(N-d)}{d} B'_{d,N}(k) \sim F(2d, 2N - 2d) . \tag{17}$$

According to the PDF of the $F$ distribution, the PDF of $B'_{d,N}(k)$ can be obtained as

$$f_{B'_{d,N}(k)}(x) = \begin{cases} \frac{1}{B(d,N-d)} \frac{x^{d-1}}{(1+x)^N} & , x > 0 \\ 0 & , x \leq 0 \end{cases} , N \geq d+1 . \tag{18}$$

where $B(d, N-d) = \frac{\Gamma(d)\Gamma(N-d)}{\Gamma(N)}$ is the beta function.

2.4.2. Probability Distribution for $B_{d,N}(k)$

The PDF of $B_{d,N}(k)$ can be determined by taking the derivative of the PDF of $B'_{d,N}(k)$:

$$f_{B_{d,N}(k)}(x) = \frac{1}{(1-x)^2} f_{B'_{d,N}(k)}\left(\frac{x}{1-x}\right) \ , 0 < x < 1 . \tag{19}$$

Substituting Equation (18) into (19) produces:

$$f_{B_{d,N}(k)}(x) = \begin{cases} \frac{x^{d-1}(1-x)^{N-d-1}}{B(d,N-d)} & ,0 < x < 1 \\ 0 & , x \le 0 \ or \ x \ge 1 \end{cases} , \ N \ge d+1 . \tag{20}$$

The probability density plot for $B_{d,N}(k)$ is shown in Figure 1.

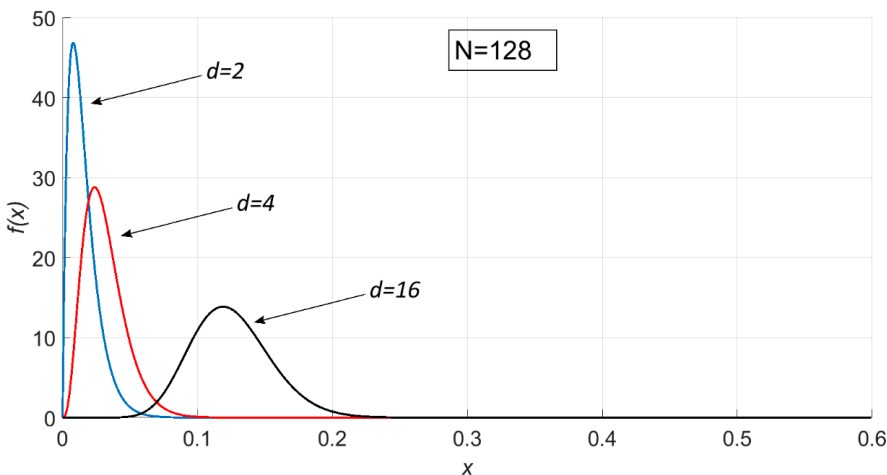

**Figure 1.** Probability density plot for $B_{d,N}(k)$, where $N = 128$ and $d$ = 2, 4, and 16.

According to the PDF of the beta distribution, $B_{d,N}(k)$ follows a beta distribution with parameters $d, N - d$, i.e.,

$$B_{d,N}(k) \sim \beta(d, N - d) . \tag{21}$$

The CDF for $B_{d,N}(k)$ is

$$F_{B_{d,N}(k)}(x) = \begin{cases} I_x(d, N - d) & ,0 < x < 1 \\ 0 & , x \le 0 \ or \ x \ge 1 \end{cases} , N \ge d+1 . \tag{22}$$

where $I_p(a, b)$ is the incomplete beta function.

It can be shown that the PDF and CDF of the PSER for GWN have no relationship to the noise variance and are only related to the number of power spectrum bins in a sub-band $d$ and the number of bins in total power spectrum $N$. The expectation and variance of the beta distribution $B_{d,N}(k)$ are then given by

$$E(B_{d,N}(k)) = \frac{d}{N} , \tag{23}$$

$$Var(B_{d,N}(k)) = \frac{Nd - d^2}{N^2(N+1)} . \tag{24}$$

*2.5. Statistical Characteristics of the PSER under $H_1$*

2.5.1. Probability Distribution for $B'_{d,N}(k)$

The term $B'_{d,N}(k)$ can be represented as

$$B'_{d,N}(k) = \frac{X'_k}{\sum_{i=0}^{N-1} X_i - X'_k} = \frac{2/\sigma^2 X'_k}{2/\sigma^2 \left(\sum_{i=0}^{N-1} X_i - X'_k\right)}, \tag{25}$$

and its numerator follows a non-central chi-square distribution with $2d$ degrees of freedom and a non-centrality parameter $\lambda'_k$, and its denominator follows a non-central chi-square distribution with $2N - 2d$ degrees of freedom and a non-centrality parameter $\sum_{i=0}^{N-1} \lambda_i - \lambda'_k$. Therefore, $B'_{d,N}(k)$ is a ratio of two non-central chi-square distributions, which is referred to a G distribution.

Let $\delta_1 = \lambda'_k/2$, $\delta_2 = \left(\sum_{i=0}^{N-1} \lambda_i - \lambda'_k\right)/2$. According to the definition of the G distribution [26], we deduce that the PDF of $B'_{d,N}(k)$ as

$$\begin{aligned} f_{B'_{d,N}(k)}(x) &= f(x; 2d, 2N - 2d; \delta_1, \delta_2) \\ &= e^{-(\delta_1+\delta_2)} \sum_{j=0}^{\infty} \sum_{l=0}^{\infty} \frac{\delta_1^j \delta_2^l}{j!l!} [B(d+j, N-d+l)]^{-1} \frac{x^{d+j-1}}{(1+x)^{N+j+l}}, \ x \geq 0. \end{aligned} \tag{26}$$

The CDF of $B'_{d,N}(k)$ is

$$F_{B'_{d,N}(k)}(x; 2d, 2N - 2d; \delta_1, \delta_2) = e^{-(\delta_1+\delta_2)} \sum_{j=0}^{\infty} \sum_{l=0}^{\infty} \frac{\delta_1^j \delta_2^l}{j!l!} I_{x/(1+x)}(j+d, N+l-d). \tag{27}$$

2.5.2. Probability Distribution for $B_{d,N}(k)$

The term $B_{d,N}(k)$ can be represented as

$$B_{d,N}(k) = \frac{X'_k}{\sum_{i=0}^{N-1} X_i} = \frac{2/\sigma^2 X'_k}{2/\sigma^2 \sum_{i=0}^{N-1} X_i}, k = 0, 1, \cdots, N' - 1. \tag{28}$$

Both the numerator and denominator of $B_{d,N}(k)$ correspond to non-central chi-square distributions. The numerator is included in the denominator; therefore, $X'_k$ and $\sum_{i=0}^{N-1} X_i$ are not independent. According to the definition of the doubly non-central beta distribution [26], $B_{d,N}(k)$ follows a doubly non-central beta distribution with parameters $2d, 2N - 2d$ and non-centrality parameters $\lambda'_k$, and $\sum_{i=0}^{N-1} \lambda_i - \lambda'$, which can be denoted as

$$B_{d,N}(k) \sim \beta_{2d,2N-2d}(\lambda'_k, \sum_{i=0}^{N-1} \lambda_i - \lambda'_k). \tag{29}$$

According to Equation (16), the CDF of $B_{d,N}(k)$ can be derived from the CDF of $B'_{d,N}(k)$:

$$F_{B_{d,N}(k)}(x) = \begin{cases} e^{-(\delta_1+\delta_2)} \sum_{j=0}^{\infty} \sum_{l=0}^{\infty} \frac{\delta_1^j \delta_2^l}{j!l!} I_x(j+d, N+l-d) & , 0 < x < 1 \\ 0 & , x \leq 0 \text{ or } x \geq 1 \end{cases}. \tag{30}$$

The derivative of Equation (30) is the PDF of $B_{d,N}(k)$:

$$f_{B_{d,N}(k)}(x) = \begin{cases} e^{-(\delta_1+\delta_2)} \sum_{j=0}^{\infty} \sum_{l=0}^{\infty} \frac{\delta_1^j \delta_2^l}{j!l!} \frac{(1-x)^{N+l-d-3} x^{j+d-1}}{B(j+d, N-d+l)} & , 0 < x < 1 \\ 0 & , x \leq 0 \text{ or } x \geq 1 \end{cases}. \tag{31}$$

Since $\delta_1$ and $\delta_2$ have relations with the noise variance $\sigma^2$, $B_{d,N}(k)$ is therefore affected by noise. Both the PDF and CDF of $B_{d,N}(k)$ are represented by infinite double series, so their

values can only be obtained through numerical computation. When $N = 128$, $d = 2, 4, 16$, $\delta_1 = 5.6$, and $\delta_2 = 20$, the density probability plot of $B_{d,N}(k)$ is shown in Figure 2.

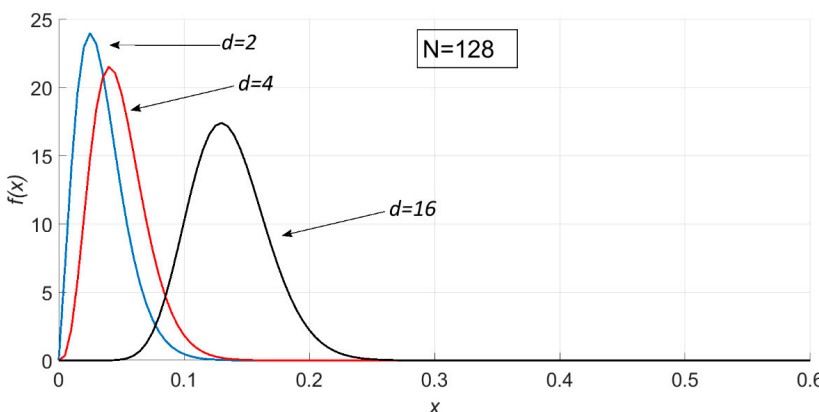

**Figure 2.** Probability density plot for $B_{d,N}(k)$, where $N = 128$, $\delta_1 = 5.6$, $\delta_2 = 20$, and $d = 2, 4$, and 16.

2.5.3. Relationship between $\delta_1 + \delta_2$ and Signal-to-Noise Ratio (SNR)

$\delta_1$ is the ratio of the energy of valid signal contained in the selected band to the noise energy of one spectral bin. $\delta_2$ is the ratio of the energy of the valid signal not contained in the selected band to the noise energy of one spectral bin.

The total power spectrum energy for the signal and noise can be represented, respectively, as

$$\sum_{i=0}^{N-1} P_X(i) = \frac{1}{N} \sum_{i=0}^{N-1} \left[ X_R^2(k) + X_I^2(k) \right] = \frac{1}{N} \sum_{i=0}^{N-1} \left( a_k^2 + b_k^2 \right) \tag{32}$$

$$\sum_{i=0}^{N-1} P_Z(i) = \frac{1}{N} \sum_{i=0}^{N-1} \left[ Z_R^2(k) + Z_I^2(k) \right] \approx \frac{1}{N} \sum_{i=0}^{N-1} N\sigma^2 = N\sigma^2 \tag{33}$$

As such, the relationship between $\delta_1 + \delta_2$ and the SNR is given by

$$
\begin{aligned}
SNR &= 10 * \log_{10}\left( \sum_{i=0}^{N-1} P_X(i) / \sum_{i=0}^{N-1} P_Z(i) \right) \\
&\approx 10 * \log_{10}\left( \sum_{k=0}^{N-1} \left( a_k^2 + b_k^2 \right) / N^2\sigma^2 \right) \\
&= 10 * \log_{10}\left( \frac{\delta_1 + \delta_2}{N} \right),
\end{aligned} \tag{34}
$$

i.e.,

$$\delta_1 + \delta_2 = N * 10^{\frac{SNR}{10}} . \tag{35}$$

Equation (35) means that when $N$ and SNR are fixed, the sum of $\delta_1$ and $\delta_2$ is also fixed.

Letting the energy of the valid signal in the entire power spectrum $E$ be the energy of the sub-band $E_1$ and the total energy of other spectral bins $E_2$; then $\delta_1 = E_1 / (N\sigma^2)$, $\delta_2 = E_2 / (N\sigma^2)$. The relationship between total energy and sub-band energy can be expressed as $E_1 = aE$, where $a$ denotes the sub-band energy ratio coefficient. It is easy to obtain $a = \delta_1 / (\delta_1 + \delta_2)$.

An indicator to measure the signal quality of the local frequency band, which is used in later text, is the local SNR. It is defined as the ratio of local frequency energy to local noise energy, i.e.,

$$\gamma' = E_1 / \left( dN\sigma^2 \right) = \delta_1 / d . \tag{36}$$

### 3. Signal Detection Based on the PSER

*3.1. Principle*

The signal detection based on the PSER takes $B_{d,N}(k)$ as the test statistic to determine whether signals are present in the sub-band of interest according to the value of $B_{d,N}(k)$. From Equations (21) and (29), we obtain

$$B_{d,N}(k) \sim \begin{cases} \beta(d, N-d) & H_0 \\ \beta_{2d,2N-2d}(\lambda'_k, \sum_{i=0}^{N-1} \lambda_i - \lambda'_k) & H_1 \end{cases}.$$

The common performance measures are the probability of false alarm $P_f$, and the probability of detection $P_d$. Under the Neyman–Pearson criteria (i.e., CFAR criterion), likelihood ratio yields the optimal hypothesis testing solution and performance are measured by a resulting pair of detection and false alarm probabilities ($P_d$ and $P_f$). Each pair is associated with the particular threshold $\eta_{PSER}$ that tests the decision statistic. If the test statistic is larger than $\eta_{PSER}$, the signal is deemed to be present, and it is absent otherwise, i.e.,

$$\begin{cases} B_{d,N}(k) < \eta_{PSER} & H_0 \\ B_{d,N}(k) \geq \eta_{PSER} & H_1 \end{cases}.$$

The false alarm probability $P_f$ can be expressed as follows:

$$\begin{aligned} P_f &= \Pr(B_{d,N}(k) \geq \eta_{PSER}|H_0) = 1 - I_{\eta_{PSER}}(d, N-d) \\ &\Leftrightarrow \eta_{PSER} = I^{-1}(1 - P_f; d, N-d) \end{aligned} \tag{37}$$

where $I^{-1}(1 - P_f; d, N-d)$ is the inverse function of $I_{\eta_{PSER}}(d, N-d)$, which $I^{-1}(x; a, b)$ can be solved by referring to the method provided in [27,28]. According to Equation (30), the detection probability $P_d$ can be expressed as follows:

$$\begin{aligned} P_d &= \Pr(B_{d,N}(k) \geq \eta_{PSER}|H_1) \\ &= 1 - e^{-(\delta_1+\delta_2)} \sum_{j=0}^{\infty} \sum_{l=0}^{\infty} \frac{\delta_1^j \delta_2^l}{j! l!} I_{\eta_{PSER}}(j+d, N+l-d) \end{aligned} \tag{38}$$

The decision domain of the PSER is shown in Figure 3.

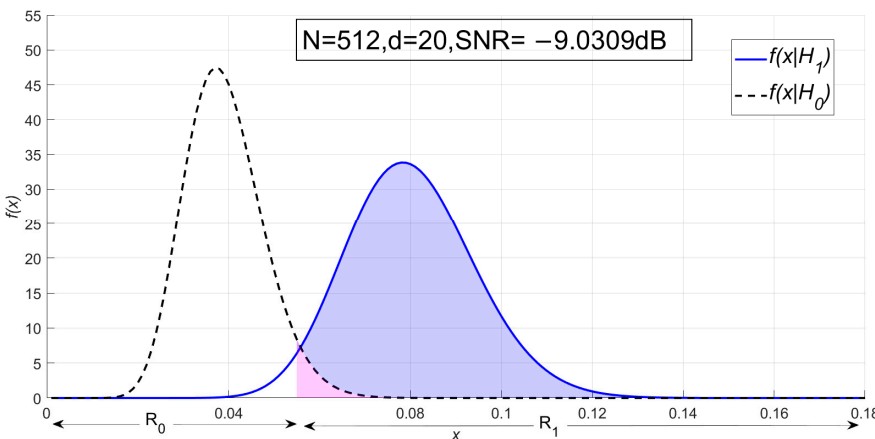

**Figure 3.** Probability density function of the power-spectrum sub-band energy ratio (PSER) and division of decision domain.

In Figure 3, $\delta_1 = 26$, $\delta_2 = 38$, and the SNR is $-9.03$ dB. The dotted line is the probability density function under $H_0$, and the solid line is the probability density function under $H_1$. $R_0$ is the decision domain of signal absence, the range of which is $x \in (0, \eta_{PSER}]$. $R_1$ is the decision domain of signal presence, the range of which is $x \in (\eta_{PSER}, 1]$.

### 3.2. Performance Comparison of the PSER with Other Energy Detection Methods

Under the same conditions, the PSER will compare the detection probability with the commonly used time-domain energy detection (TDED) [29] and local spectral energy detection (LSED) methods to evaluate the detection performance of the PSER.

#### 3.2.1. Time-Domain Energy Detection

If the value of the signal at time $n$, $x(n)$, is treated as a constant, then

$$s(n) = x(n) + z(n) \sim \mathcal{N}\left(x(n), \sigma^2\right).$$

The decision metric for the energy detector $T_{TD}$ can be written as follows:

$$T_{TD} = \sum_{n=1}^{N} s^2(n). \tag{39}$$

Therefore, $T_{TD}/\sigma^2$ obeys a non-central chi-square distribution with $N$ degrees of freedom and a non-centrality parameter $\lambda$; that is,

$$\frac{T_{TD}}{\sigma^2} \sim \chi_N^2(\lambda), \ \lambda = \frac{1}{\sigma^2} \sum_{n=1}^{N} |x(n)|^2.$$

Letting SNR $\gamma = \frac{1}{N^2\sigma^2} \sum\limits_{n=1}^{N} |x(n)|^2$, then the entire distribution of $T_{TD}$ in the TDED model is described as follows:

$$T_{TD} \sim \begin{cases} \mathcal{N}(N\sigma^2, 2N\sigma^4) & H_0 \\ \mathcal{N}((1+\gamma)N\sigma^2, (1+2\gamma)2N\sigma^4) & H_1 \end{cases}, \tag{40}$$

Under the CAFR strategy, the false alarm probability $P_f$ and detection probability $P_d$ can be expressed as follows:

$$P_f = \Pr(T_{TD} \geq \eta_{TD}|H_0) = Q\left(\frac{\eta_{TD} - N\sigma^2}{\sqrt{2N}\sigma^2}\right), \tag{41}$$

$$P_d = \Pr(T_{TD} > \eta_{TD}|H_1) = Q\left(\frac{\eta_{TD} - (1+\gamma)N\sigma^2}{\sqrt{(1+2\gamma)2N}\sigma^2}\right), \tag{42}$$

$\eta_{TD}$ can be derived from Equation (41),

$$\eta_{TD} = \sqrt{2N}Q^{-1}(P_f)\sigma^2 + N\sigma^2, \tag{43}$$

and, by substituting Equation (43) into Equation (42), $P_d$ can then be evaluated as follows:

$$P_d = Q\left(\frac{Q^{-1}(P_f) - \gamma\sqrt{\frac{N}{2}}}{\sqrt{1+2\gamma}}\right). \tag{44}$$

#### 3.2.2. Local-Spectrum Energy Detection

The detection object of LSED is also the sub-band of interest; therefore, LSED is the method closest to the PSER. If the sub-band is composed of $P(i)$, $P(I+1)$, ..., $P(i+d-1)$ and total $d$ spectral bins, the detection statistic can be defined as follows:

$$T_{LFD} = \sum_{l=i}^{i+d-1} P(l).$$

According to the central limit theorem (CLT), the distribution of $T_{LFD}$ is approximately

$$T_{LFD} \sim \begin{cases} \mathcal{N}\left(d\sigma^2, d\sigma^4\right) & H_0 \\ \mathcal{N}\left((1+\gamma')d\sigma^2, (1+2\gamma')d\sigma^4\right) & H_1 \end{cases}, \tag{45}$$

where $\gamma'$ is the local SNR, as denoted in Equation (36). The false probability $P_f$ is

$$P_f = \Pr(T_{LFD} \geq \eta_{LFD}) = Q\left(\frac{\eta_{LFD} - d\sigma^2}{\sqrt{d}\sigma^2}\right), \tag{46}$$

Then, $\eta_{LFD}$ can be derived from Equation (46),

$$\eta_{LFD} = \sqrt{d}\sigma^2 Q^{-1}(P_f) + d\sigma^2. \tag{47}$$

The detection probability $P_d$ can be expressed as follows:

$$P_d = \Pr(T_{LFD} > \eta_{LFD}|H_1) = Q\left(\frac{Q^{-1}(P_f) - \gamma'\sqrt{d}}{\sqrt{1+2\gamma'}}\right). \tag{48}$$

### 3.2.3. Theoretical Detection Probabilities Comparison

In this section, the detection performance of the PSER, LSED, and TDED is compared, and the effects of sub-band energy ratio coefficient $a$, local bin number $d$, and full spectrum bin number $N$ on the detection performance of the PSER are discussed.

When $N = 512$, $d = 20$, $P_f = 0.01$, and $a = 1$ (i.e., the sub-band contains all the energy of signal), the theoretical detection probabilities of the three methods are shown in Figure 4.

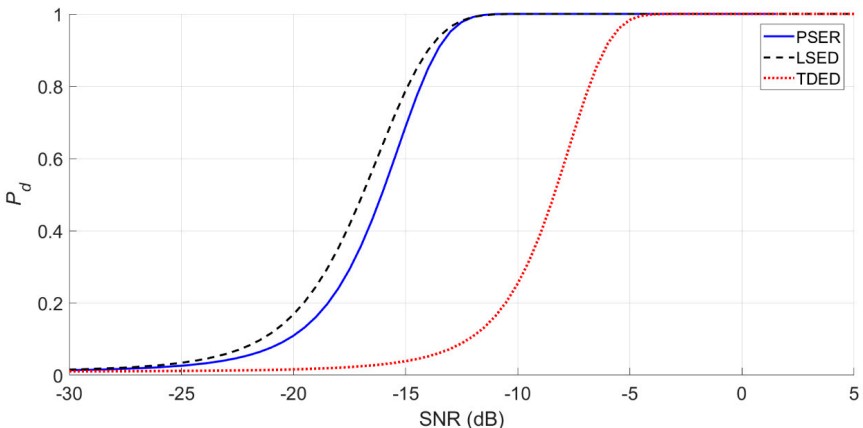

**Figure 4.** Detection probabilities of the three methods compared when the sub-band contains all the energy ($N = 512$, $d = 20$, and $P_f = 0.01$).

It can be seen from Figure 4 that the detection probability of the PSER is better than that of TDED, but it is slightly worse than that of LSED.

When the sub-band energy ratio coefficient changes, the detection probabilities of the PSER and LSED are as follows.

In Figure 5, the smaller the sub-band energy ratio coefficient is, the worse the detection probabilities of the PSER and LSED are. When $a = 1$, the detection probabilities of LSED and the PSER were the best. When $a = 0.2$, the detection probabilities of LSED and the PSER decrease, and even the detection probability of the PSER is inferior to that of TDED.

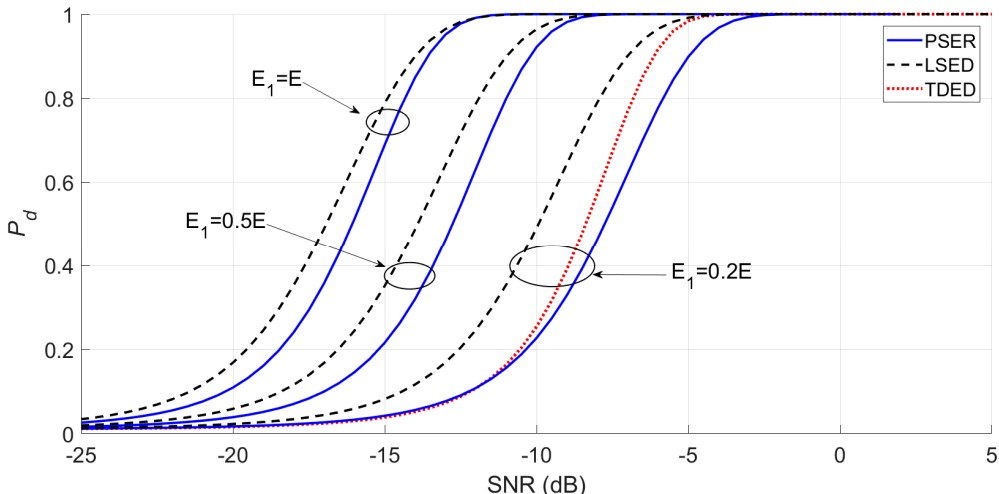

**Figure 5.** Detection probabilities of the PSER and local spectral energy detection (LSED) when the sub-band energy ratio coefficient varies ($N = 512$, $d = 20$, $P_f = 0.01$, and $a = 1, 0.5, 0.2$).

When the sub-band bin number $d$ changes, the detection probabilities of the PSER and LSED are as follows.

In Figure 6, the smaller the sub-band bin number is, the better the detection probabilities of the PSER and LSED are. However, the detection probability of the PSER is always worse than that of LSED.

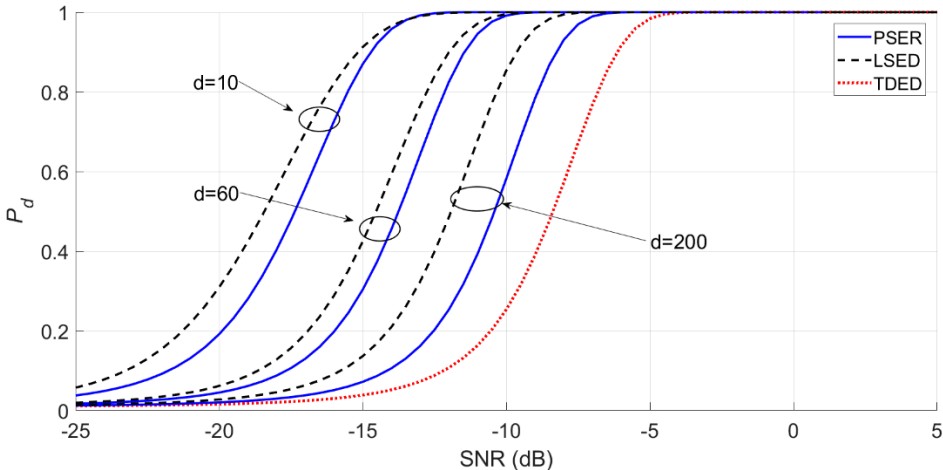

**Figure 6.** Detection probabilities of the PSER and LSED when the sub-band bin number varies ($N = 512$; $d = 10, 60$, and 200; $P_f = 0.01$; and $a = 1$).

When the full spectrum bin number $N$ changes, the detection probabilities of the PSER and LSED are as follows.

In Figure 7, the greater the full spectrum bin number $N$ is, the better the detection probabilities of the PSER and LSED are. However, the detection probability of the PSER is slightly worse than that of LSED.

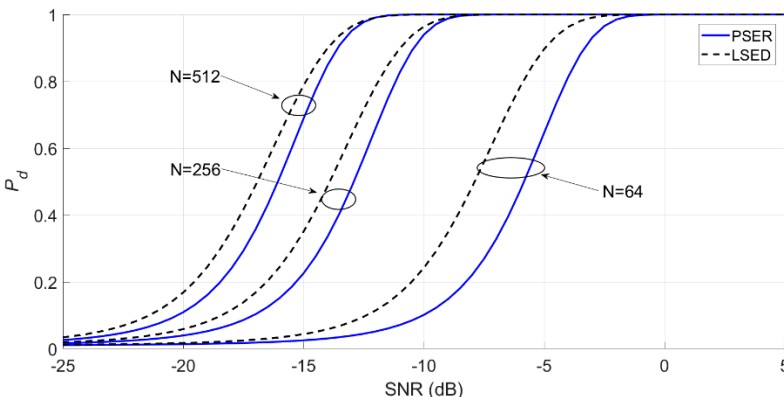

**Figure 7.** Detection probabilities of the PSER and LSED when the full spectrum bin number varies ($N$ = 64, 256, and 512; $d$ = 20; $P_f$ = 0.01; $a$ = 1).

The detection probability of the PSER is influenced by the sub-band energy ratio coefficient $a$, local bin number $d$, and full spectrum bin number $N$. When $d$ decreases, $N$ increases, or $a$ increases, the detection performance of the PSER will improve. In general, the detection performance of the PSER is lower than that of LSED but is generally higher than that of TDED. However, the detection performance of the PSER will be lower than that of TDED when the sub-band contains less signal energy.

### 3.3. Detection Performance of the PSER under Noise Uncertainty

In an actual environment, there are many electromagnetic disturbances, including GWN, quantized noise, and thermal noise. Therefore, the noise intensity is time-varying, the noise is only approximately Gaussian distribution, and the noise power fluctuates within a certain range, which is called the noise power uncertainty (NPU) interval. The NPU interval can be expressed as $\sigma^2 \in \left[\sigma_l^2, \sigma_h^2\right]$, where $\sigma_l^2$ and $\sigma_h^2$ are the upper and lower bounds of noise power, respectively. When $\sigma^2 = \sigma_h^2$, the noise power is very high and the detector may think there is an effective signal, thus causing a false alarm error. When $\sigma^2 = \sigma_l^2$, the actual noise power is much lower than the noise power estimated by the receiver, which makes the detector fail to detect the effective signal and causes the missed detection error.

Tandra et al. [30] presented a method to quantitatively describe the NPU interval by the NPU coefficient. The NPU coefficient is denoted $\rho \geq 0$, and the nominal noise power is denoted $\sigma_n^2$. Letting $\rho = 10\lg(\sigma_h^2/\sigma_n^2) = 10\lg(\sigma_n^2/\sigma_l^2)$; then, $\sigma_l^2 = 10^{-\rho/10}\sigma_n^2$ and $\sigma_h^2 = 10^{\rho/10}\sigma_n^2$. The NPU interval can be expressed as $\sigma^2 \in \left[10^{-\rho/10}\sigma_n^2, 10^{\rho/10}\sigma_n^2\right]$. In general, noise power obeys a uniform distribution in the NPU interval; that is, $\sigma^2 \sim U\left(\sigma_l^2, \sigma_h^2\right)$.

To make the system achieve the ideal detection performance, $\sigma^2$ should then maximize $P_f$ and minimize $P_d$, which is called the robust statistics approach (RSA) in the literature [30].

#### 3.3.1. Detection Performance of TDED under Noise Uncertainty

Under the RSA, we used the upper-bound value $\sigma_h^2$ of $\sigma^2$ to calculate the false alarm probability and the lower-bound value $\sigma_l^2$ of $\sigma^2$ to calculate the detection probability; that is,

$$T_{TD} \sim \begin{cases} \mathcal{N}\left(N\sigma_h^2, 2N\sigma_h^4\right) & H_0 \\ \mathcal{N}\left((1+\gamma)N\sigma_l^2, (1+2\gamma)2N\sigma_l^4\right) & H_1 \end{cases}. \tag{49}$$

The false alarm probability $P_f$ can be expressed as

$$P_f = \Pr(T_{TD} \geq \eta_{TD}|H_0) = Q\left(\frac{\eta_{TD} - N\sigma_h^2}{\sqrt{2N}\sigma_h^2}\right) \tag{50}$$

$\eta_{TD}$ can be derived from Equation (50).

$$\eta_{TD} = \sqrt{2N}\sigma_h^2 Q^{-1}(P_f) + N\sigma_h^2 .$$

The detection probability $P_d$ can be expressed as

$$P_d = \Pr(T_{TD} > \eta_{TD}|H_1) = Q\left(\frac{\eta_{TD}-(1+\gamma)N\sigma_l^2}{\sqrt{2N(1+2\gamma)\sigma_l^2}}\right) = Q\left(\frac{Q^{-1}(P_f)10^{\frac{\varrho}{5}}+\left(10^{\frac{\varrho}{5}}-1-\gamma\right)\sqrt{\frac{N}{2}}}{\sqrt{1+2\gamma}}\right) \quad (51)$$

### 3.3.2. Detection Performance of LSED under Noise Uncertainty

According to the RSA criterion, the upper-bound value $\sigma_h^2$ and lower-bound value $\sigma_l^2$ of $\sigma^2$ are used to calculate the false alarm probability and missing detection probability

$$T_{LFD} \sim \begin{cases} \mathcal{N}\left(d\sigma_h^2, d\sigma_h^2\right) & H_0 \\ \mathcal{N}\left((1+\gamma')d\sigma_l^2, (1+2\gamma')d\sigma_l^4\right) & H_1 \end{cases} . \quad (52)$$

The false alarm probability $P_f$ and detection probability $P_d$ can be expressed as

$$P_f = \Pr(T_{LFD} \geq \eta_{LFD}) = Q\left(\frac{\eta_{LFD}-d\sigma_h^2}{\sqrt{d}\sigma_h^2}\right), \quad (53)$$

$$P_d = \Pr(T_{LFD} \geq \eta_{LFD}|H_1) = Q\left(\frac{\eta_{LFD}-(1+\gamma')d\sigma_l^2}{\sqrt{(1+2\gamma')d\sigma_l^2}}\right) = Q\left(\frac{10^{\frac{\varrho}{5}}Q^{-1}(P_f)+\sqrt{d}\left(10^{\frac{\varrho}{5}}-1-\gamma'\right)}{\sqrt{1+2\gamma'}}\right) \quad (54)$$

### 3.3.3. Detection Performance of PESR under Noise Uncertainty

Equation (37) shows that the PSER is not affected by noise variance under noise uncertainty; that is, the value of false alarm probability $P_f$ is independent of noise. The detection probability of the PSER is

$$P_d = 1 - e^{-(\delta_1+\delta_2)}\sum_{j=0}^{\infty}\sum_{l=0}^{\infty}\frac{\delta_1^j\delta_2^l}{j!l!}I_{\eta_{PSER}}(j+d, N+l-d) .$$

If $P_d$ is to be the smallest, $e^{-(\delta_1+\delta_2)}\sum_{j=0}^{\infty}\sum_{l=0}^{\infty}\frac{\delta_1^j\delta_2^l}{j!l!}I_{\eta_{PSER}}(j+d, N+l-d)$ should be the largest. Since $e^{-(\delta_1+\delta_2)}\sum_{j=0}^{\infty}\sum_{l=0}^{\infty}\frac{\delta_1^j\delta_2^l}{j!l!}I_{\eta_{PSER}}(j+d, N+l-d)$ is an increasing function of $\sigma^2$, $\sigma^2$ should therefore be $\sigma_h^2$. Letting $\delta'_1 = 10^{-\rho/10}\delta_1$ and $\delta'_2 = 10^{-\rho/10}\delta_2$, then

$$\begin{aligned}
P_d &= \Pr(B_{d,N}(k) \geq \eta_{PSER}|H_1) \\
&= 1 - e^{-(\delta'_1+\delta'_2)}\sum_{j=0}^{\infty}\sum_{l=0}^{\infty}\frac{\delta'^j_1\delta'^l_2}{j!l!}I_{\eta_{PSER}}(j+d, N+l-d) \\
&= 1 - e^{-10^{\frac{-\rho}{10}}(\delta_1+\delta_2)}\sum_{j=0}^{\infty}\sum_{l=0}^{\infty}\frac{10^{\frac{-\rho}{10}(j+l)}\delta_1^j\delta_2^l}{j!l!}I_{\eta_{PSER}}(j+d, N+l-d) .
\end{aligned} \quad (55)$$

### 3.3.4. Detection Performance Comparison of Three Methods under Noise Uncertainty

When $N$ = 512, $d$ = 20, $a$ = 1, and $\rho$ = 0.5 and 1, the influence on the detection probabilities of the PSER, LSED, and TDED are shown in Figures 8 and 9.

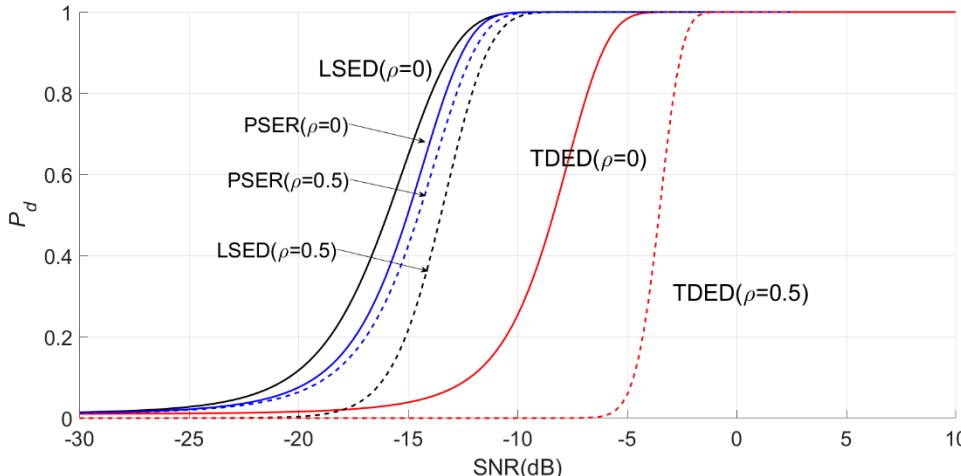

**Figure 8.** Effects of noise uncertainty on the PSER, LSED, and time-domain energy detection (TDED), where $\rho$ = 0.5.

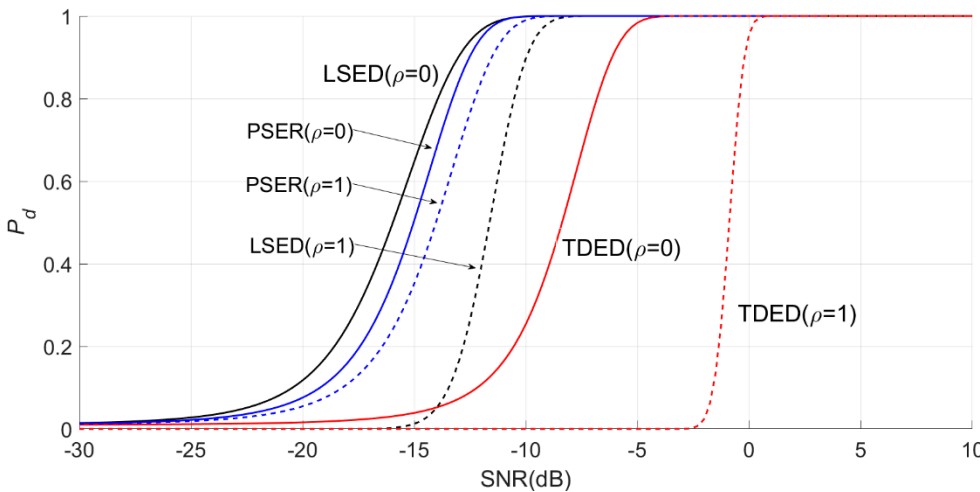

**Figure 9.** Effects of noise uncertainty on the PSER, LSED, and TDED, where $\rho$ = 1.

In Figures 8 and 9, the solid line is the detection probability under certain noise (i.e., $\rho = 0$), and the dotted line is the detection probability under noise uncertainty, i.e., $\rho > 0$. Under certain noise, the detection probability of LSED is the best, followed by that of the PSER, while that of TDED is the worst. When $\rho$= 0.5, the detection probability of TDED decreases the most, followed by that of LSED, while that of the PSER is decreased only slightly. When $\rho$= 1, the detection probabilities of three methods decrease more significantly than that of $\rho$= 0.5, but the decrease amplitude of the PSER is the smallest. Obviously, in the case of noise power uncertainty, the PSER is less affected by noise uncertainty, and its robustness is better than that of LSED and TDED, which is the largest advantage of the PSER in signal detection.

## 4. Experiments

For this section, we verified and compared the detection performances of the PSER, LSED, and TDED methods discussed in Section 3 through Monte Carlo simulations and the vibration signal collected by an optical time domain reflectometer (OTDR).

### 4.1. Simulations

We used cosine signals to simulate narrow-band signals, which could ensure all the energy contained in the sub-band by using narrow-band signals. In this case, the energy proportion coefficient was 1.

We used Ricker wavelet signals to simulate broadband signals because the energy occupied by the local frequency band is easy to control. In this case, the energy proportion coefficient was less than 1.

We performed all Monte Carlo simulations for at least $10^5$ independent trials. We set $P_f$ to 0.01 and $N$ to 512. We used mean-square error (MSE) to measure the deviation between the theoretical values and actual statistical results.

### 4.1.1. Narrow-Band Signal

We used the cosine signal mixed with GWN as experimental data. The amplitude of the sinusoidal signal was 0.5, the frequency was 30 Hz, the phase was 0, and the duration was 1 s. For convenience of calculation, we set the frequency resolution as 1 Hz. We used a bilateral spectrum in the experiment. Because the energy in the power spectrum of the signal was concentrated at approximately 30 Hz, the selected frequency band was 21–42 Hz; that is, it contained 22 spectral bins. When the SNR was $-10$ dB, one cosine signal and its power spectral density (PSD), under noise absence and presence, respectively, are shown in Figure 10.

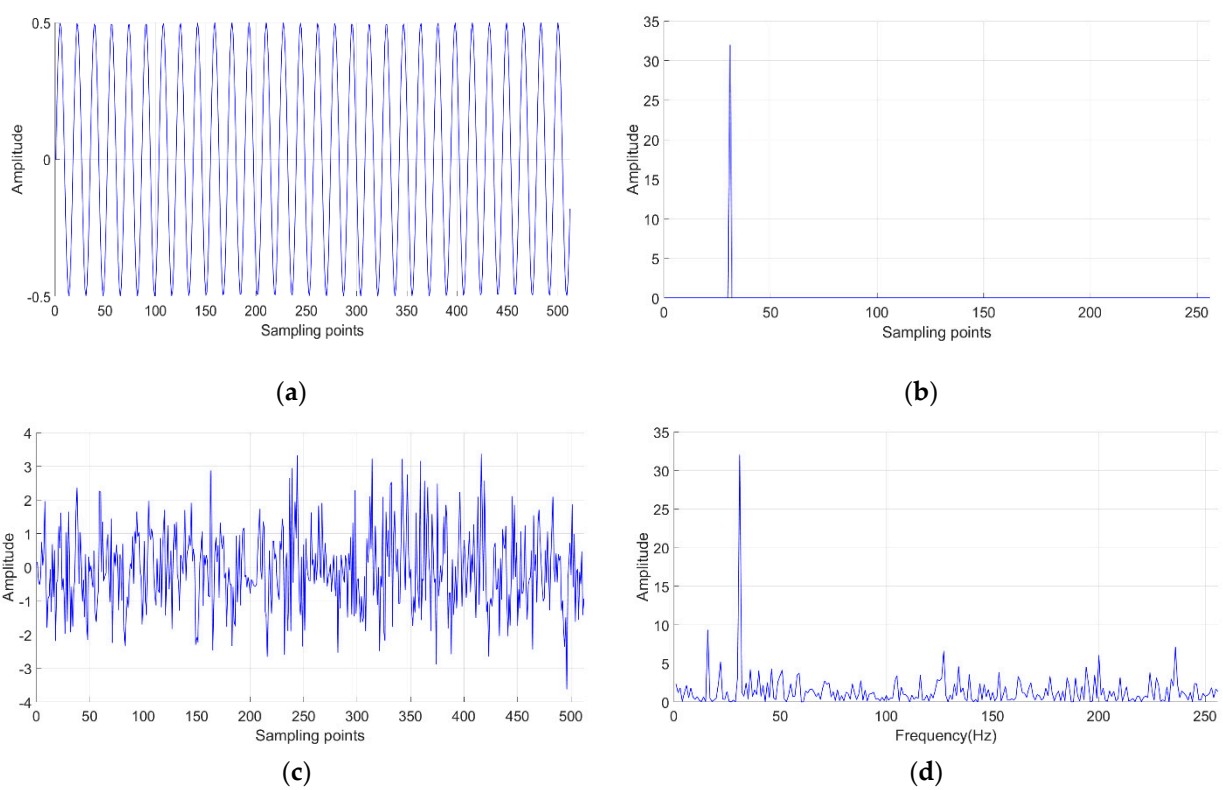

**Figure 10.** Cosine signal and its power spectrum under noise absence and presence. (**a**) Waveform in time domain without noise. (**b**) Power spectrum of (**a**). (**c**) Waveform in time domain with noise, where SNR is –10 dB. (**d**) Power spectrum of (**c**).

When $d$ was 22, $\eta_{PSER} = I^{-1}(0.99; 22, 490) = 6.6\%$ and the sub-band 21–42 Hz contained 8.7% of all energy. When $d$ is 5, $\eta_{PSER} = I^{-1}(0.99; 5, 507) = 2.3\%$, and, the sub-band 29–33 Hz contained 5.22% of all energy. The false alarm probabilities and detection probabilities obtained by the three methods are as shown in Figures 11 and 12.

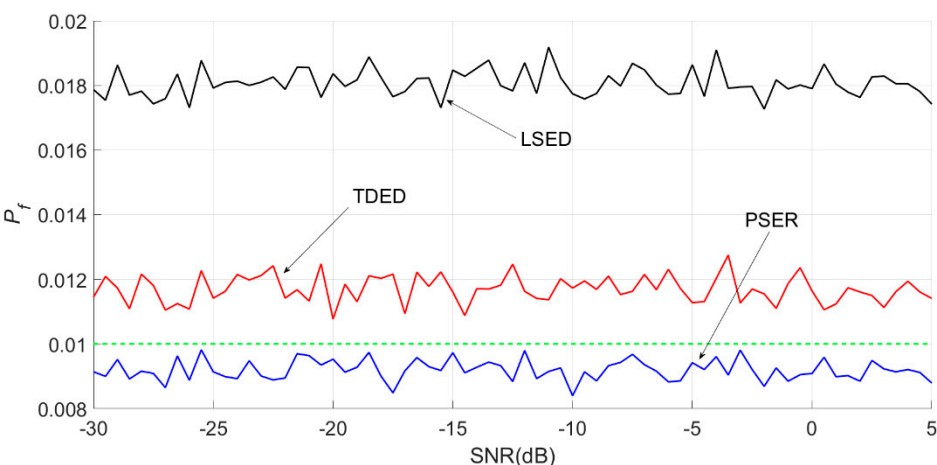

**Figure 11.** Actual false alarm probabilities obtained by three methods for narrow-band signals when $N = 512$ and $d = 22$. The green dotted line is the pre-set theoretical false alarm probability.

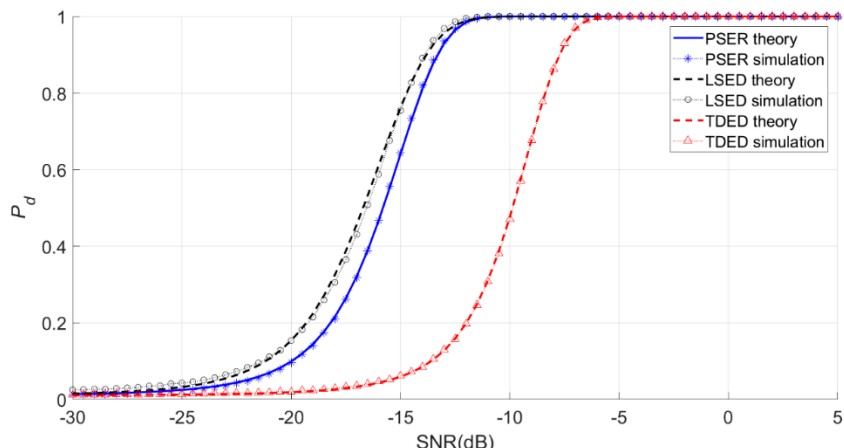

**Figure 12.** Theoretical and actual detection probabilities obtained by three methods for narrow-band signals when $N = 512$.

The average false alarm probability of the PSER was 0.0091, that of LSED was 0.0181, and that of the time-domain energy method was 0.0117. Therefore, as shown in Figure 11, the actual false alarm probability of the PSER was below 0.01, that of TDED was above 0.01, and that of LSED was close to 0.02.

As seen in Figure 12, when the sub-band energy ratio coefficient was good, e.g., $a = 1$, the detection performance of the PSER was better than that of TDED, while the detection performance of LSED was better than that of the PSER.

The MSEs of the actual and theoretical probabilities of these experiments are given in Table 1.

**Table 1.** Mean square errors (MSEs) between actual and theoretical probabilities for narrow-band signals.

| Probability | PSER | LSED | TDED |
|:---:|:---:|:---:|:---:|
| $P_f$ | $0.0075 \times 10^{-4}$ | $0.6535 \times 10^{-4}$ | $0.0308 \times 10^{-4}$ |
| $P_d$ | $0.0615 \times 10^{-4}$ | $0.7531 \times 10^{-4}$ | $0.0441 \times 10^{-4}$ |

It can be seen from Table 1 that the MSE between the actual and theoretical false alarm probabilities of the PSER was the lowest, followed by that of TDED and LSED. At the same time, the MSE between the actual and theoretical detection probabilities of TDED was the smallest, that of the PSER was slightly higher than that of TDED, and that of LSED

was significantly higher. The deviation between the PSER actual and theoretical detection probabilities was very small, which indicated that the PSER method is accurate.

### 4.1.2. Broadband Signal

The PSER is often used for vibration monitoring, especially in microseismic detection, while the Ricker wavelet is often used to simulate seismic or microseismic signals in underground vibration simulation experiments. Therefore, the Ricker wavelet was selected as the simulation broadband signal in this paper. The expression for its time domain is

$$s(t) = \left(1 - 2\pi^2 f_M^2 t^2\right) e^{-\pi^2 f_M^2 t^2} , \tag{56}$$

where $t$ is time and $f_M$ represents the central frequency. A Fourier transform is applied to Equation (56) to obtain the expression of the Ricker wavelet in the frequency domain:

$$S(f) = \frac{2f^2}{\sqrt{\pi} f_M^3} e^{-\frac{f^2}{f_M^3}}. \tag{57}$$

The Ricker wavelet used in this experiment had a central frequency of 50 Hz. Its time-domain waveform and power spectrum waveform under noise absence and presence, respectively, are shown in Figure 13.

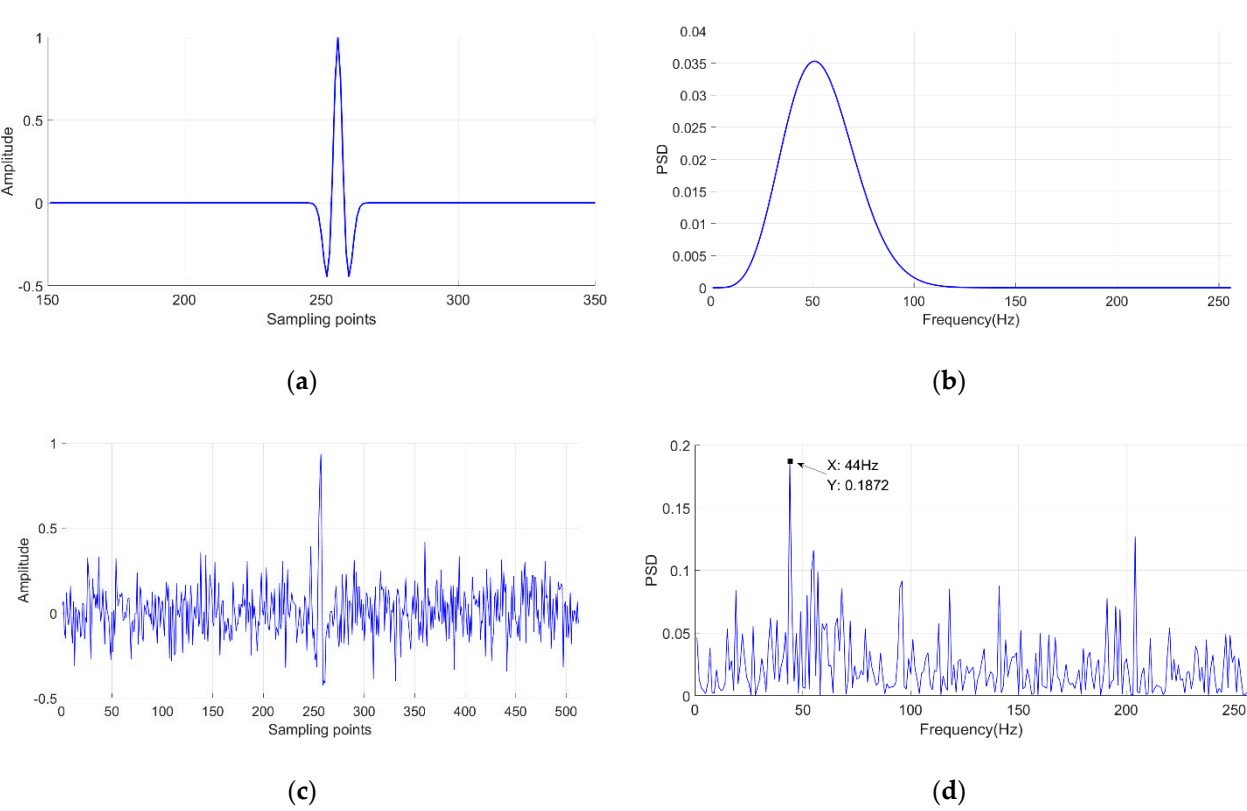

**(a)**

**(b)**

**(c)**

**(d)**

**Figure 13.** Oscillogram of Ricker wavelet under no noise and noise. $f_M$ = 50 Hz. (**a**) Waveform in time domain without noise. (**b**) Power spectrum of (**a**). (**c**) Waveform in time domain with noise, where SNR is −5 dB. (**d**) Power spectrum of (**c**), and there is a peak on 44 Hz.

The experiment was focused on the two cases of high-local SNR and poor-local SNR to highlight the influences of $\delta_1$ and $\delta_2$ on the PSER.

High-Local SNR

When we set the sub-band of interest at 36–65 Hz, the frequency band occupied approximately 61.5% of the total energy (i.e., $a = 0.615$). The theoretical and actual detection probabilities of the three methods are shown in Figures 14 and 15.

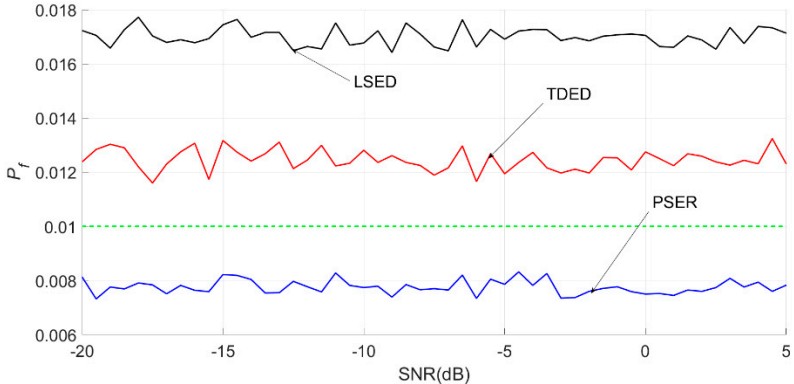

**Figure 14.** Actual false alarm probabilities obtained by three methods for the Ricker wavelet with local bandwidths of 36–65 Hz.

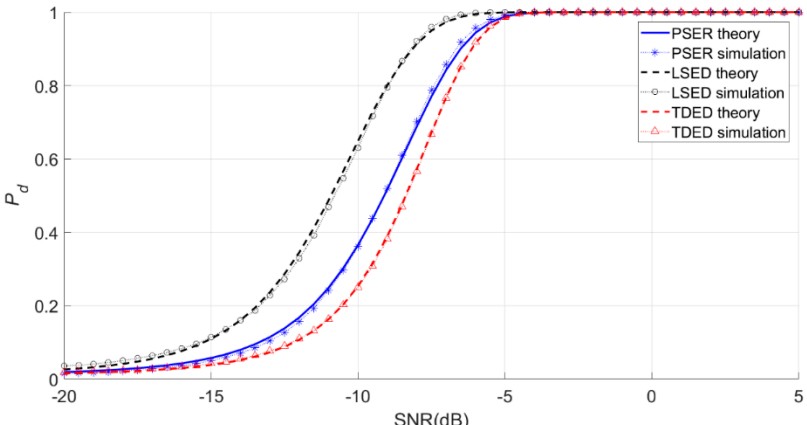

**Figure 15.** Actual and theoretical detection probabilities obtained by three methods for the Ricker wavelet with local bandwidths of 36–65 Hz.

In Figure 15, it is shown that the detection performance of the PSER was better than that of TDED, while the detection performance of LSED was better than that of the PSER.

Low-Local SNR

We set the sub-band of interest at 21–40 Hz, and the frequency band occupied only approximately 21% of the total energy (i.e., $a = 0.21$). The theoretical and actual detection probabilities of the five methods are shown in Figures 16 and 17.

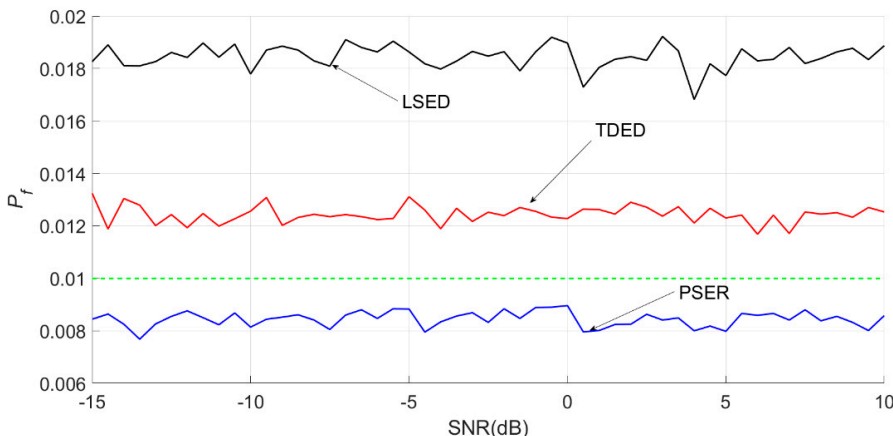

**Figure 16.** Actual false alarm probabilities obtained by three methods for the Ricker wavelet with local bandwidths of 21–40 Hz.

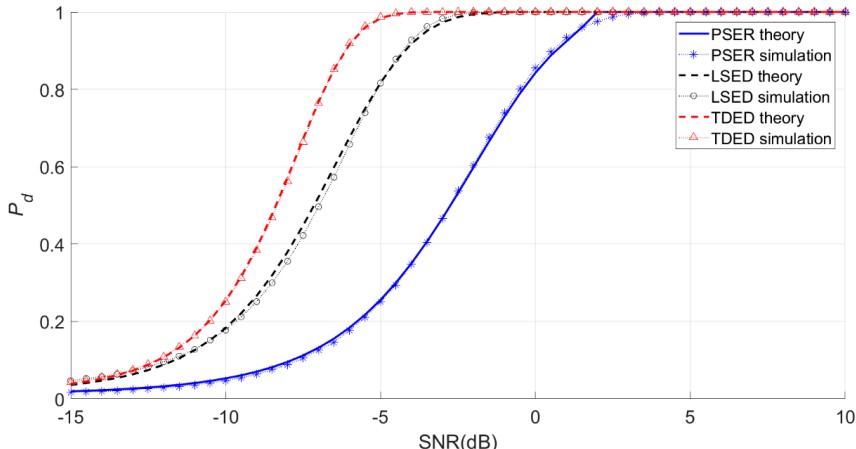

**Figure 17.** Actual and theoretical detection probabilities obtained by three methods for the Ricker wavelet with local bandwidths of 21–40 Hz.

Figure 17 shows that the detection probabilities of the PSER and LSED were lower than that of TDED when the energy proportion coefficient was low. Therefore, the sub-band energy ratio coefficient should not be too low to ensure the good detection performance of the PSER.

Comparison of Mean Square Error

The MSEs of the actual and theoretical probabilities of these experiments are given in Table 2.

**Table 2.** MSEs between actual and theoretical probabilities for the Ricker wavelet.

| Bandwidth (Hz) | Probability | PSER | LSED | TDED |
|---|---|---|---|---|
| 35–65 | $P_f$ | $0.0508 \times 10^{-4}$ | $0.4921 \times 10^{-4}$ | $0.0614 \times 10^{-4}$ |
| | $P_d$ | $0.5019 \times 10^{-4}$ | $0.6254 \times 10^{-4}$ | $0.0886 \times 10^{-4}$ |
| 21–40 | $P_f$ | $0.0250 \times 10^{-4}$ | $0.7161 \times 10^{-4}$ | $0.0603 \times 10^{-4}$ |
| | $P_d$ | $0.4483 \times 10^{-4}$ | $0.9455 \times 10^{-4}$ | $0.0801 \times 10^{-4}$ |

As shown in Table 2, whether under high or low local SNR, the MSE between the actual and theoretical false alarm probabilities of the PSER was the smallest and that of TDED was second. The MSE between the actual and theoretical detection probabilities of TDED was the biggest, that of the PSER was the second, and that of LSED was the smallest.

Therefore, the difference between the actual and theoretical detection probabilities of PSER was not large, which indicates that the detection method based on PSER was accurate for broadband signals.

### 4.1.3. Vibration Signal Detection

The vibration signals were collected by an NBX-S3000 instrument (Nebrex Co., Hyogo, Japan). The vibration signals were generated by a tapping machine, which is a standard piece of vibratory equipment. There were five percussion hammers on the tapping machine, and the tapping interval was 0.1 s. Photographs of the NBX-S3000 instrument and tapping machine are shown in Figure 18.

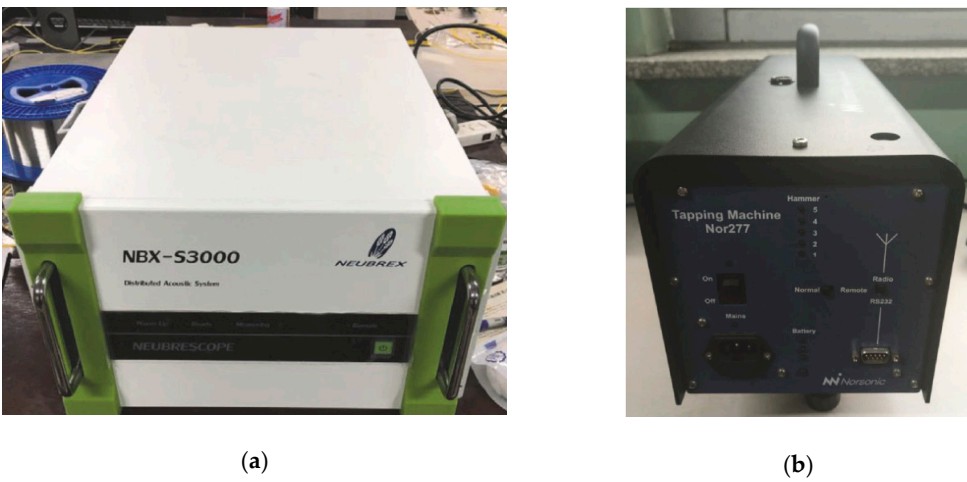

(**a**)                                     (**b**)

**Figure 18.** (**a**) NBX-S3000 and (**b**) vibratory equipment.

The OTDR technique monitors the vibration sensed by an optical fiber. The shape of an optical fiber is a line. One fiber can be divided into many segments of equal length. Each segment is equivalent to a sensor, and the length is denoted as spatial resolution, i.e., the minimum space distance that an OTDR instrument can discern.

The state of one fiber with one vibration source is shown in Figure 19. According to the vibration attenuation law, the vertical intersection point between the vibration source and optical fiber is the strongest point of vibration.

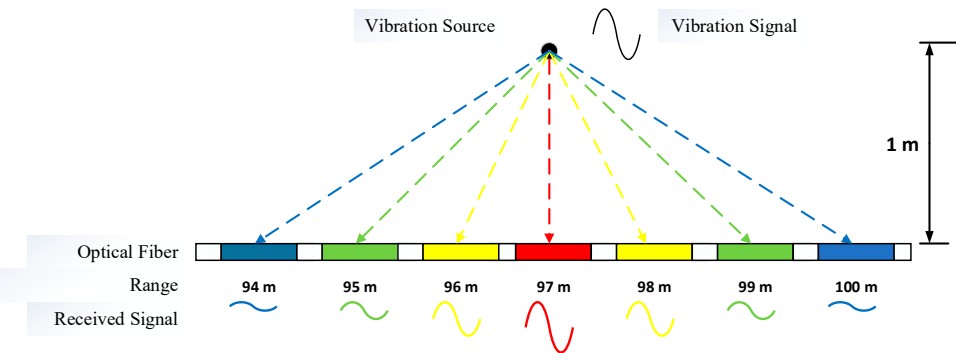

**Figure 19.** Diagram of optical fiber receiving vibration signal.

In this experiment, the fiber was buried approximately 10 cm below ground. The spatial resolution was 1 m, the sampling rate 4000 Hz, and the monitoring range was from 66 to 126 m from the start side. The tapping machine was fixed almost 97 m from the start side and 1 m from the fiber.

#### 4.1.3.1. Background Noise

Before detecting the vibration signal, the background noise collected by the NBX-S3000 instrument, totaling 600,000 records, were analyzed. We found that the noise at different distances had different statistical properties, such as mean, variance, and power spectrum.

Figure 20 shows the mean and variance at different distances. The average PSD values at 97 m and 85 m, which were calculated by the Welch method, are shown in Figure 21.

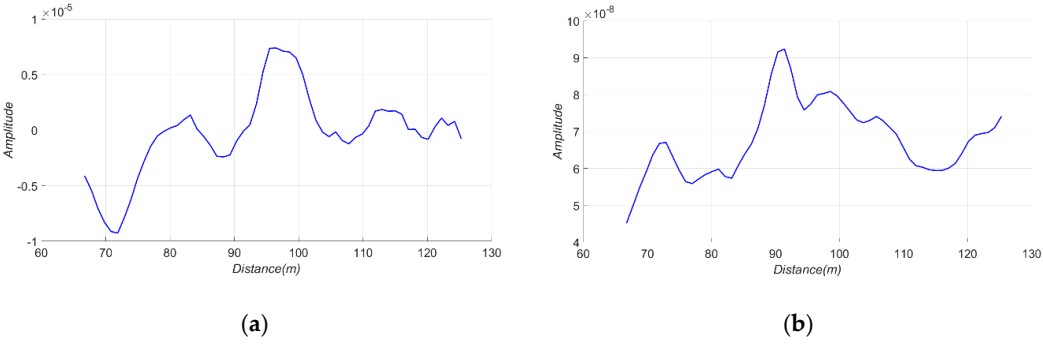

(**a**)          (**b**)

**Figure 20.** (**a**) Mean and (**b**) variance at different distances.

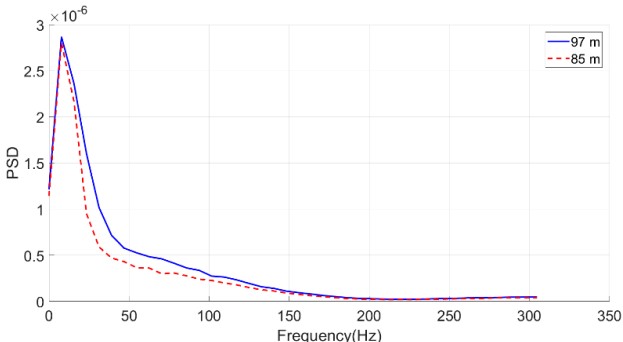

**Figure 21.** Average power spectral density (PSD) at 97 m and 85 m.

It can be seen that the background noise was not true GWN; rather, it was more like pink noise.

#### 4.1.3.2. Tapping Signal

The recording time of the tapping signal was 10 s, and the tapping interval was 0.1 s; therefore, 100 tapping samples were taken at one distance. The time-domain waveform and PSD of the tapping signal at 97 and 85 m are shown in Figures 22 and 23, respectively.

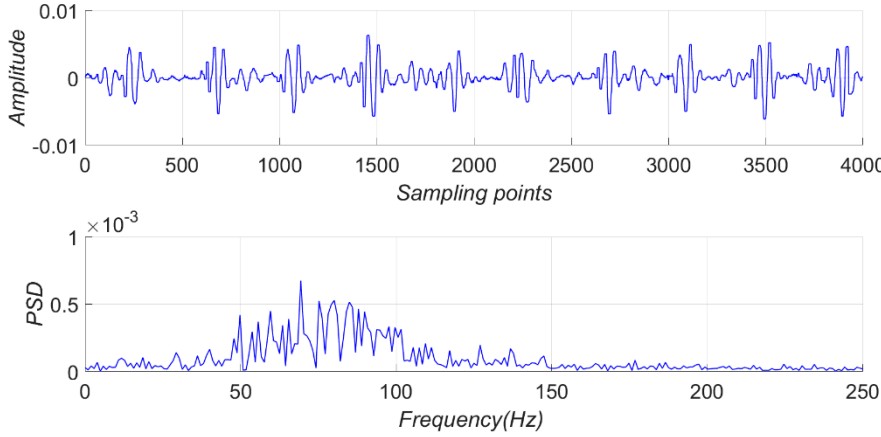

**Figure 22.** Time domain waveform and PSD of the tapping signal at 97 m.

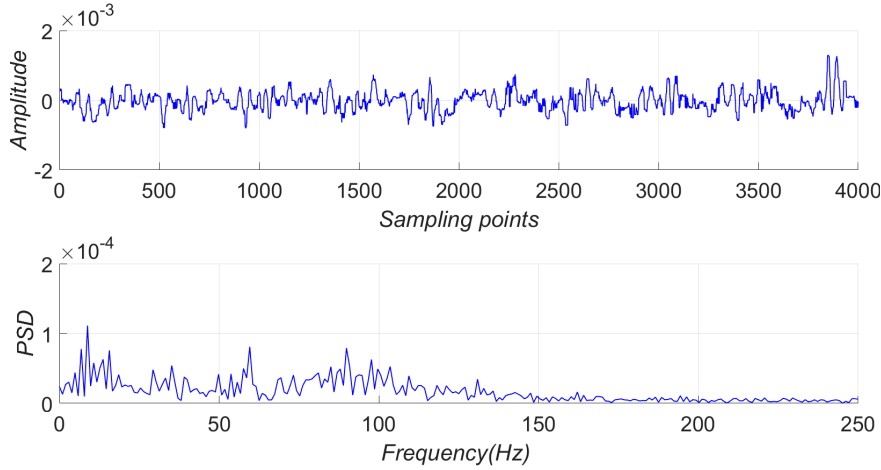

**Figure 23.** Time domain waveform and PSD of the tapping signal at 85 m.

The tapping signal at 97 m was easier to discern than that at 85 m in the time domain. The bins at 97 m between 50 Hz and 100 Hz were higher than the other bins. Therefore, the characteristic frequency of tapping was in the range of 50–100 Hz. The bins at 85 m between 50 Hz and 100 Hz seemed to be higher than other bins, but it was not obvious.

The approximate SNRs from 84 m to 108 m are shown in Figure 24. The signal at 97 m was closest to the vibration source, so it had a high SNR. The signal at 85 m or 108 m was far from the vibration source, so it had a low SNR.

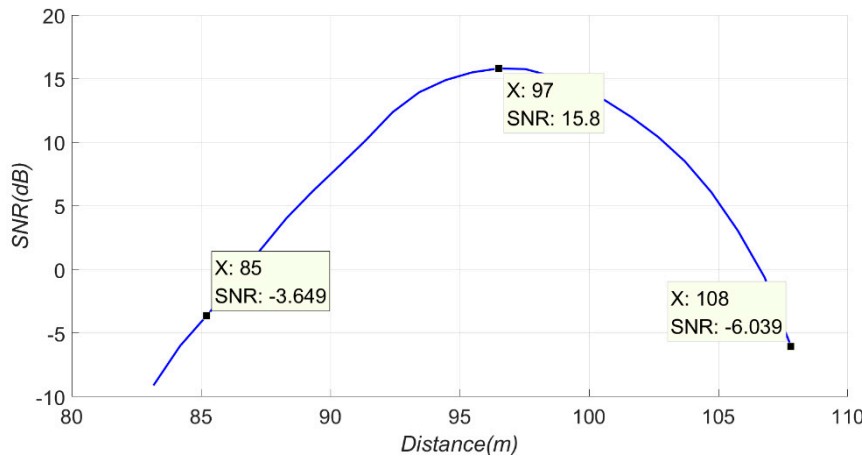

**Figure 24.** Approximate SNR from 84 to 108 m.

4.1.3.3. Detection

Since the noise in the tapping signal was not GWN, we had to change the colored noise into white noise before using LSED and the PSER. We obtained a new PSD by subtracting the average PSD of noise from the PSD of the tapping signal. The new PSD was used in the LSED and the PSER.

When the sub-band was 66–105 Hz, $N$ was 1024, $P_f$ was 0.05, and $d$ was 11. The false alarm and detection probabilities of the PSER, LSED, and TDED are shown in Figures 25 and 26.

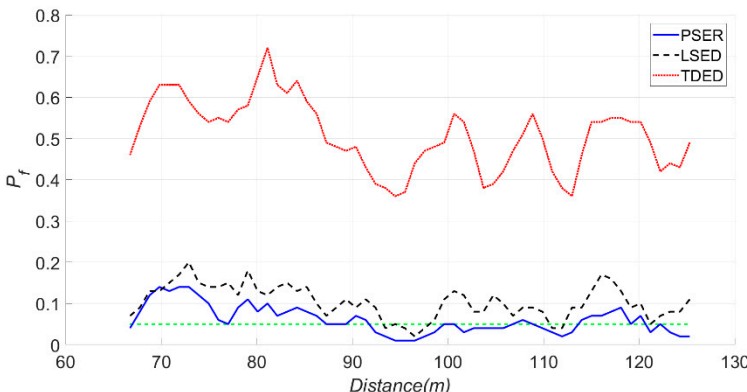

**Figure 25.** False alarm probabilities of the PSER, LSED, and TDED, where the sub-band was 66–105 Hz, $N$ was 1024, and $P_f$ was 0.05.

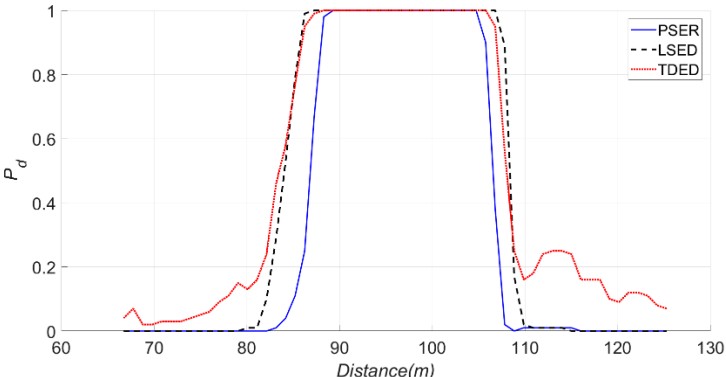

**Figure 26.** Detection probabilities of the PSER, LSED, and TDED, where the sub-band was 66–105 Hz, $N$ was 1024, and $P_f$ was 0.05.

The false alarm probability of TDED was approximately 60%, which indicates that indicated that the detection result of TDED was incorrect.

As seen in Figure 26, the detection probability of LSED was greater than that of the PSER, and the false alarm probabilities of LSED and the PSER were close to *a*, which indicated that the detection results of LSED and the PSER were correct.

Under noise uncertainty, when $\rho$ was 5, the change can be seen in Figure 27.

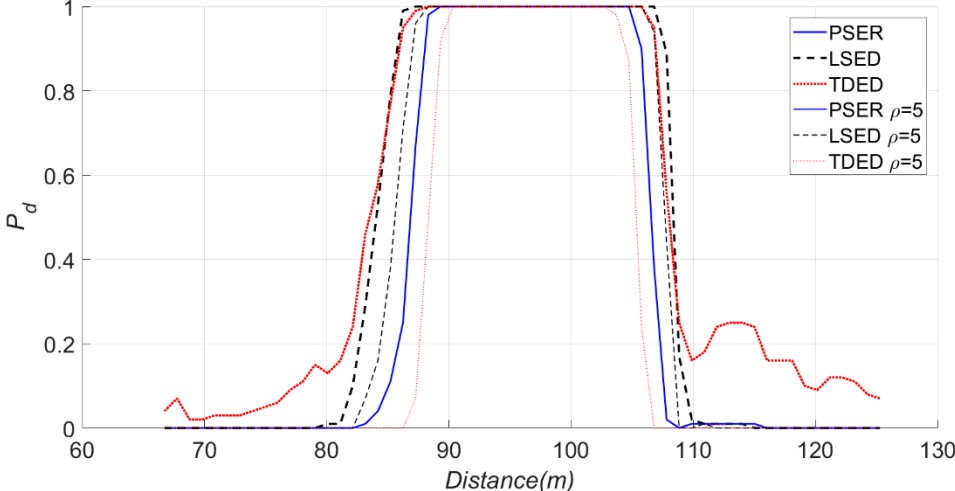

**Figure 27.** Effects of noise uncertainty on the PSER, LSED, and TDED, where $\rho$ = 5.

In Figure 27, the thick lines are the detection probabilities of three methods under original noise and the thin lines are the detection probabilities under noise uncertainty. The detection probabilities of LSED and TDED were both decreasing; however, that of the PSER did not change markedly; that is to say, the robustness of the PSER was better than that of LSED and TDED.

## 5. Discussion

### 5.1. Rationale for Rectangular Window Function Selection

In spectral analysis, it is necessary to add a symmetric window function to a sampling sequence in order to suppress side-lobe interference and improve accuracy. This symmetric window function can be expressed as

$$w_N(k) = a - (1-a)\cos(2\pi k/N), k = 0, 1, \cdots, N-1 . \tag{58}$$

A value of $a = 1$ corresponds to a rectangular window, $a = 0.5$ indicates a Hanning window, and $a = 0.54$ implies a Hamming window. In this study, a rectangular window was used to calculate the spectrum with Equation (2). A rectangular window exhibits inherent limitations, but unlike other window functions, it can ensure that spectral bins remain mutually independent after a discrete Fourier transform of the signal. The use of other window functions often produces a correlation between the power spectrum bins, which makes any subsequent calculation of probability distributions extremely difficult.

### 5.2. Rationality of CDF for the PSER

The expression of CDF for the PSER appearing in the previous literature [2] was greatly improved in this paper, especially when the number of spectral bins in the sub-band was relatively small, such as $d < 10$. In [2], the numerator and denominator of the random variable $B'_{d,N}(k)$ were regarded as obeying the Gaussian distribution, and its CDF was

$$F_{B'_{d,N}(k)}(z) = \frac{1}{2} + \frac{1}{2}erf\left(\frac{\mu_B z - \mu_A}{\sqrt{2(\sigma_B^2 z^2 + \sigma_A^2)}}\right) . \tag{59}$$

This approximation is reasonable if the number of bins in the numerator and denominator is large enough. However, if the number of bins in the numerator is small, this approximation is subject to a large deviation. In this paper, the numerator and denominator of $B'_{d,N}(k)$ were regarded as non-central chi-square distributions. When the number of spectral bins in the numerator was relatively small, the cumulative distribution function was still very accurate.

### 5.3. Calculation of CDF for the PSER

The CDF for the PSER derived in this paper was expressed by infinite double series, and its value could only be obtained by numerical calculation. Since the number of calculation terms was set to be large, it took a significant amount of calculation time. However, the CDF for the PSER is only used in theoretical analysis rather than being required in actual detection. How to improve the calculation speed of CDF needs further research.

### 5.4. Detection Performance of the PSER Depends on Sub-Band Energy Ratio Coefficient

The experiment in Section 4 showed that the detection performance of the PSER depends on the sub-band energy ratio coefficient when the SNR is fixed, and the higher the sub-band energy ratio coefficient is, the better the detection performance will be. However, further study is needed to determine how much the sub-band energy ratio coefficient can guarantee that the detection performance of the PSER is better than that of TDED.

### 5.5. Advantages of the PSER

The PDF and CDF of the PSER under GWN have nothing to do with the variance of noise; rather, they are only related to the number of spectral bins of the entire spectrum and sub-band. This means that a detector based on the PSER does not need to estimate noise variance. However, TDED and LSED need to estimate noise variance, and if the variance is estimated incorrectly, the detection performances of TDED and LSED will decrease greatly. Therefore, the detection performance of the PSER is better than that of TDED and LSED under noise uncertainty.

### 6. Conclusions

In this paper, a systematic investigation of the statistical characteristics for power-spectrum sub-band energy ratios in the cases of signal absence and signal presence is presented. The statistical characteristics of the PSER provided a theoretical foundation for the use of the PSER in signal detection. The results demonstrated that the PSER of signal absence and signal presence followed beta and doubly non-central beta distributions, respectively. According to the statistical characteristics of the PSER, a signal detection method based on the PSER was established. We found that the detection performance of this method was inferior to that of LSED and generally superior to that of TDED. However, the robustness of PSER detection performance under noise uncertainty was the best among the three methods.

**Author Contributions:** Conceptualization, Y.H. and H.L.; formal analysis, Y.H. and H.L.; investigation, Y.H. and H.L.; methodology, Y.H. and H.L.; validation Y.H., S.W., and H.L.; writing—original draft, H.L.; writing—review and editing, Y.H. and S.W. All authors have read and agreed to the published version of the manuscript.

**Funding:** This research was supported by the Science and Technology Nova Plan of Beijing City (No. Z201100006820122) and Fundamental Research Funds for the Central Universities (No. 2020RC14).

**Institutional Review Board Statement:** Not applicable.

**Informed Consent Statement:** Not applicable.

**Data Availability Statement:** Restrictions apply to the availability of these data. Data sharing is not applicable to this article.

**Conflicts of Interest:** The authors declare no conflict of interest.

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
