# Peer review of "Signal Detection Based on Power-Spectrum Sub-Band Energy Ratio"

_electronics, doi:10.3390/electronics10010064_

Round 1
Reviewer 1 Report
The paper is very well written.
Section 2 gives a good background of PSER and its statistical characteristics, which sets up the signal detection principle described in section 3. The method compares well with other common methods (LSED & TDED) and outperforms them in the case of uncertain noise. This is significant for signal detection.
Monte-Carlo simulations presented in section 4 confirm the utility of this method as a useful tool for signal detection.
Author Response
Thank you very much for your suggestions and comments.I have carefully revised the errors in the article.
Reviewer 2 Report
In the abstract clarify “signal absence”
Line 40: is: The PSER of ,Bdn(k) is defined….is confused. Reformulate.
Line 68: There is a signal in the signal sequence: Reformulate.
Line 87 Define sigma.
In section 4 a short insight should be done on what cosine signals with noise by plotting the typical fft, or PSD’s, for each test case.. The authors spend a great deal of the explaining the theoretical bases of PSER, and very short time about explaining the test signals, which are an important step. Pf in figure 10 is not mentioned in the text
Caption of figure 10 is very poor. Should be improved. The false alarm probabilities seem to be independent of the
SNR, for each method. Comment comparing with figure 11.
Tests should be performed for selected real-life signals such as biomedical signals (ECG, EEG etc) added to artificial controllable noise.
It should be commented the feasibility of the method on the case of signal detection on non-stationary signal time series. Is this method realistically applicable to these cases?
Author Response
Point 1: In the abstract clarify “signal absence”
Response 1: “signal absence” means that there is only noise and no signal. We have changed “signal absence” to “pure noise”.
Point 2: Line 40: is: The PSER of ,Bdn(k) is defined….is confused. Reformulate.
Response 2: We have changed “ The PSER of ,Bdn(k) is defined” to “ The PSER Bdn(k) is defined”
Point 3: Line 68: There is a signal in the signal sequence: Reformulate.
Response 3: We have changed “ There is a signal in the signal sequence” to “ There is a signal on the sampling sequence”
Point 4: Define sigma.
Response 4: Sigma is the variance of noise, and it has been defined in Line 35.
Point 5: In section 4 a short insight should be done on what cosine signals with noise by plotting the typical fft, or PSD’s, for each test case. The authors spend a great deal of the explaining the theoretical bases of PSER, and very short time about explaining the test signals, which are an important step.
Response 5: In section 4, we added many figures on test signal. These figures include original signal, original PSD, signal with noise, and the PSD with noise. Figure 10 and figure 13 was new figures.
Point 6: Pf in figure 10 is not mentioned in the text. Caption of figure 10 is very poor. Should be improved. The false alarm probabilities seem to be independent of the SNR, for each method. Comment comparing with figure 11.
Response 6: In CAFR, no matter how SNR changes, Pf should be a constant. The Figure 10 has changed to Figure 11. If the Pf in Figure 11 is not a straight line or deviate too much from the preset threshold, then these detection methods perhaps are wrong.
Point 7: Tests should be performed for selected real-life signals such as biomedical signals (ECG, EEG etc) added to artificial controllable noise.
It should be commented the feasibility of the method on the case of signal detection on non-stationary signal time series. Is this method realistically applicable to these cases?
Response 7: In the revised paper, we added the detection on vibration signal which were collected by optical time domain reflectometer (OTDR). The detection performance on vibration signal conformed to the conclusion obtained in the paper. When noise variance estimation was accurate, detection probability of LESD was better than PSER, however, that of TDED was better than that of LSED for the affection of the colored noise. In the case of variance estimation error, the detection probabilities of LSED and TDED decreased significantly, however, that of PSER did not decrease much.
Reviewer 3 Report
The manuscript is written poorly that makes it difficult to conduct a proper scientific evaluation. The reviewer feels a simple signal detection method based on power-spectrum sub-band energy ratio do not offer much new information. The PSER already used in many applications especially audio classification. Major comments on technical approaches and ideas: The literature review is very weak. The limitation of other methods need to be address clearly. As the authors also agreed that there are several methods for signal detection under noisy environment. Therefore, the novelty of the manuscript should be highlighted. The section 3 was also poorly written. It has to rewritten again. Definitions of various parameters and applied calculation algorithms are not clear, and, should be explained more deeply. Many symbols in the mathematical formulas are either not correctly explained in the text or are not explained at all. The reviewer doubts about originality of figure 12. It is not allowed on such reputed journal. The reviewer also expected to see the method is working on real environment. Therefore, the authors should consider some case study. The reviewer suggest that the authors should go through the entire manuscript to remove inconsistencies and add clarification as needed.Author Response
Point 1: The reviewer feels a simple signal detection method based on power-spectrum sub-band energy ratio do not offer much new information. The PSER already used in many applications especially audio classification. Major comments on technical approaches and ideas: The literature review is very weak.
Response 1: The PSER method has been used for decades, but its long history does not mean that its theoretical basis is well studied. There is no exact conclusion on which probability distribution PSER obeys to. The existing expression of PSER's cumulative distribution function is very complex or inaccurate. The expression in this paper has not appeared in other literatures, and it is very accurate and can be used in the case that the number of bin is very small. The detector based PSER in this paper is not appeared in other literatures too.
Point 2: The limitation of other methods need to be address clearly. As the authors also agreed that there are several methods for signal detection under noisy environment. Therefore, the novelty of the manuscript should be highlighted.
Response 2: PSER is one Energy detection method. Therefore, we compared PSER with two commonly used energy detection methods, LSED and TDED. The detection performance of PSER is not the best, but it is least affected by noise uncertainty, and there is no need to estimate noise variance before using PSER. LSED has the best detection performance, but it needs to estimate noise variance before use, and it is susceptible to uncertain noise. TDED has the worst detection performance, it also estimates noise variance before use, and it is susceptible to uncertain noise.
Point 3: The section 3 was also poorly written. It has to rewritten again. Definitions of various parameters and applied calculation algorithms are not clear, and, should be explained more deeply. Many symbols in the mathematical formulas are either not correctly explained in the text or are not explained at all.
Response 3: We have add the explanations of mathematical symbols in section 3. Many mathematical parameters are explained in section 2. Because there are many mathematical symbols in this paper, I understand the trouble of not finding the explanation of the symbols. But I can ensure that all the parameters are explained when they appear. For example, λ is defined in line 103. and δ1 and δ2 are defined in line 143. σ in line 206 is defined in line 37.
Point 4: The reviewer doubts about originality of figure 12. It is not allowed on such reputed journal.
Response 4: Figure 12, which has changed to figure 13, was drown by MTALB. Maybe it was not clear enough, so we adjusted it again.
Point 5: The reviewer also expected to see the method is working on real environment. Therefore, the authors should consider some case study.
Response 5: In the revised paper, we added the detection on vibration signal which were collected by optical time domain reflectometer (OTDR). The detection performance on vibration signal conformed to the conclusion obtained in the paper. When noise variance estimation was accurate, detection probability of LESD was better than PSER, however, that of TDED was better than that of LSED for the affection of the colored noise. In the case of variance estimation error, the detection probabilities of LSED and TDED decreased significantly, however, that of PSER did not decrease much. These results can be seen in Figures 25-27. We didn't include this part at first because the original text was too long.
Point 6: The reviewer suggest that the authors should go through the entire manuscript to remove inconsistencies and add clarification as needed.
Response 6: Thanks for your suggestion, we have removed many inconsistencies and added clarification in new paper. There are too many changes to list them all.
Round 2
Reviewer 3 Report
The work that is the subject of my review concerns scientifically significant issues, and the results of the authors research may also have some applications in engineering practice. The second version of this manuscript is much better than the previous version.
However, I will be more happy to see if the authors take necessary action to improve their literature, especially between line number 60 - 69.
Please update with more recent works under a noisy environment.
Author Response
Point 1: However, I will be more happy to see if the authors take necessary action to improve their literature, especially between line number 60 - 69. Please update with more recent works under a noisy environment.
Response 1: We have added five references between line number 64 - 74. These references almost contain studies on statistical properties related to PSER so far